# Effect of Probiotic *Lactobacillus plantarum* on *Streptococcus mutans* and *Candida albicans* Clinical Isolates from Children with Early Childhood Caries

**DOI:** 10.3390/ijms24032991

**Published:** 2023-02-03

**Authors:** Yan Zeng, Ahmed Fadaak, Nora Alomeir, Yan Wu, Tong Tong Wu, Shuang Qing, Jin Xiao

**Affiliations:** 1Eastman Institute for Oral Health, University of Rochester Medical Center, Rochester, NY 14642, USA; 2Department of Biostatistics and Computational Biology, University of Rochester Medical Center, Rochester, NY 14642, USA; 3College of Dental Medicine, Columbia University, New York, NY 10032, USA

**Keywords:** *Streptococcus mutans*, *Candida albicans*, *Lactobacillus plantarum*, probiotics, multispecies biofilm

## Abstract

Probiotics interfere with pathogenic microorganisms or reinstate the natural microbiome. *Streptococcus mutans* and *Candida albicans* are well-known emerging pathogenic bacteria/fungi for dental caries. In this study, three probiotic *Lactobacilli* strains (*Lactobacillus plantarum* 8014, *L. plantarum* 14917, and *Lactobacillus salivarius* 11741) were tested on *S. mutans* and *C. albicans* clinical isolates using a multispecies biofilm model simulating clinical cariogenic conditions. The ten pairs of clinical isolates of *S. mutans* and *C. albicans* were obtained from children with severe early childhood caries. Our study findings show a remarkable inhibitory effect of *L. plantarum* 14917 on *S. mutans* and *C. albicans* clinical isolates, resulting in significantly reduced growth of *S. mutans* and *C. albicans*, a compromised biofilm structure with a significantly smaller microbial and extracellular matrix and a less virulent microcolony structure. FurTre, plantaricin, an antimicrobial peptide produced by *L. plantarum*, inhibited the growth of *S. mutans* and *C. albicans*. The mechanistic assessment indicated that *L. plantarum* 14917 had a positive inhibitory impact on the expression of *S. mutans* and *C. albicans* virulence genes and virulent structure, such as *C. albicans* hypha formation. Future utilization of *L. plantarum* 14917 and/or its antimicrobial peptide plantaricin could lead to a new paradigm shift in dental caries prevention.

## 1. Introduction

Probiotics are living bacteria found in dairy products, chewing gum, and gummies and confer health benefits on the host when delivered in adequate quantities [1,2,3]. Probiotics potentially interfere with pathogenic microorganisms or reinstate the natural microbiome [4]. Probiotics, when ingested, compete for adhesion sites of pathogens and nutrients that can co-aggregate and produce antimicrobial compounds, thus inhibiting the adhesion and the growth of pathogens on biofilm [5,6]. Probiotic *Lactobacilli* produce antimicrobial agents such as organic acids, hydrogen peroxide, and antifungal compounds such as fatty acids and bacteriocins, which have bactericidal and bacteriostatic characteristics [7].

Oral microorganisms are the most common culprits of dental caries after host and environmental factors [8]. *Streptococcus mutans* is the most widely known pathogenic bacteria causing dental caries [9], due to its acidogenicity, aciduric property, and capability of forming an extracellular matrix that comprises the main component of dental plaques/biofilms. Furthermore, studies have demonstrated the potential cariogenic role of fungi, highlighting the presence of high levels of *Candida* species in children with early childhood caries (ECC) [10,11,12,13], and *C. albicans* yields an organic acid that is believed to play a role in caries formation [14].

Studies have indicated a potential role of probiotics in managing oral health diseases, such as caries disease, periodontal disease, halitosis [15,16], and candidiasis [17,18]. Few studies have demonstrated the inhibitory effect of probiotic *Lactobacillus* on *S. mutans* and *C. albicans* in vitro [14,19,20]. Intriguingly, our previous work examined the impact of four probiotic *Lactobacillus* strains on *S. mutans* and *C. albicans* wild-type strains using a thorough multispecies biofilm model that mimicked a high-caries-risk clinical condition [21]. These tested *Lactobacillus* strains were *Lactobacillus rhamnosus* ATCC 2836, *Lactobacillus plantarum* ATCC 8014, *Lactobacillus plantarum* ATCC 14917, and *Lactobacillus salivarius* ATCC 11741 [21]. Among the examined probiotic species, *L. plantarum* and *L. salivarius* showed better growth suppression of *C. albicans* and *S. mutans* wild-type strains, disrupting the formation of virulent biofilms with fewer bacteria and exopolysaccharides (EPSs) and the development of virulent microcolonies structures [21]. Our results provide data and a rationale for further assessment of the effect of the probiotic *L. plantarum* on clinically isolated *S. mutans* and *C. albicans*, since various studies have shown that clinically isolated bacteria and yeast have a higher tolerance to antimicrobial reagents.

Hence, the objective of this study was to examine the effect of three probiotic *Lactobacillus* species (*L. salivarius* and *L. plantarum*) on the growth of clinically isolated *S. mutans* and *C. albicans* from children with severe early childhood caries (S-ECC) in multispecies planktonic and biofilm conditions and further assessed mechanistic interactions between probiotic *Lactobacillus* species and cariogenic *S. mutans* and *C. albicans.* Promising study results will lead to a new paradigm of dental caries prevention using probiotics.

## 2. Results

### 2.1. Characteristics of S-ECC Children whose S. mutans and C. albicans Were Isolated

The demographic–socioeconomic–oral health conditions of the S-ECC children the *C. albicans* and *S. mutans* isolated from are shown in Table 1. The S-ECC children had an average age of 3.5 ± 1.0 years of age, with an equal number of males and females. The majority of the children brushed their teeth daily and did not attend daycare. The average plaque index was 1.8 ± 0.6. The average decayed teeth number and decayed surface number were 11.7 ± 5.1 and 27.2 ± 17.4, respectively.

### 2.2. L. plantarum 14917 Inhibited the Growth of S. mutans and C. albicans Clinical Isolates in Planktonic Condition

The growth of *C. albicans*, *S. mutans*, and *L. plantarum* 14917 in multispecies planktonic conditions is plotted in Figure 1. *L. plantarum* inhibited the growth of *C. albicans* (Figure 1A) by <1-log after 6 h and approximately 2-log at 20 h (*p* < 0.05 at 20 h). *L. plantarum* significantly inhibited the growth of *S. mutans* (Figure 1B) at 6 h by 1-log, and completely inhibited the growth of *S. mutans* after 20 h (*p* < 0.05 at 20 h). *L. plantarum* (Figure 1C) maintained a stable growth during the 20 h of interaction with *S. mutans* and *C. albicans*. The culture medium pH (Figure 1D) dropped rapidly with the addition of *L. plantarum* in the planktonic condition, and both groups reached the same acidic level at 20 h.

Of note is that among the three *Lactobacillus* spp. tested in the screening step against *C. albicans* and *S. mutans* isolated from two S-ECC children, the inhibitory effects of *L. plantarum* 14917 and *L. salivarius* were similar, but superior to that of *L. plantarum* 8014 (See Appendix A).

### 2.3. Inhibition of C. albicans Hypha Formation by L. plantarum

The inhibition of *C. albicans*’ switching from yeast to the hyphal form was observed in the planktonic condition when treated with *L. plantarum* 14917 (Figure 2). The addition of *L. plantarum* 14917 reduced the growth of *C. albicans* and inhibited the switching from yeast to the hypha and pseudohypha form.

### 2.4. L. plantarum 14917 Inhibited Biofilm Formation by S. mutans and C. albicans Clinical Isolates

During the screening step, *L. plantarum* 14917 and *L. salivarius* 11741 were added to the biofilms formed by *S. mutans* and *C. albicans* isolated from two S-ECC children, respectively. Both *L. plantarum* 14917 and *L. salivarius* 11741 inhibited the growth of *C. albicans* and *S. mutans* (Appendix A). Although both *L. plantarum* 14917 and *L. salivarius* 11741 became the dominant species after 48 h of incubation, *L. plantarum* 14917 had a higher composition at 24 h (40%, Appendix A) compared to *L. salivarius* 11741 (20%, Appendix A). Furthermore, the growth of *L. plantarum* 14917 within the multispecies biofilms remained at a higher level than *L. salivarius* 11741 by 2-log at 72 h (Appendix A). For the reasons stated above, *L. plantarum* 14917 advanced to the assessment of impact on biofilm structure and mechanistic interaction assessment.

The growth of *C. albicans* and *S. mutans* was significantly inhibited by *L. plantarum* 14917 and is plotted in Figure 3. In the group treated with *L. plantarum* 14917, *C. albicans* was reduced by 3-log compared to the control group (Figure 3A). At 48 h, after two administrations of *L. plantarum* 14917, *S. mutans* was inhibited entirely in biofilms of the treatment group (Figure 3B), and the number of *L. plantarum* dropped during the first 24 h due to the low-sucrose (0.1%) condition and remained growing steady after 24 h in the high-sucrose (1%) condition (Figure 3C). The pH of the culture medium was significantly decreased when *L. plantarum* 14917 was added at 24, 48, and 72 h, compared to the control group (*p* < 0.05) (Figure 3D). *L. plantarum* 14917 became the dominant species after 48 h of incubation (Figure 3E). Biofilm formation was significantly reduced when *L. plantarum* 14917 was added, with a significant reduction in dry weight at 72 h (Figure 3F). In addition, the total biofilm proteins formed by the clinically isolated *S. mutans* and *C. albicans* with/without *L. plantarum* 14917 treatment were quantified. The biofilm total protein of the control group (*S. mutans* and *C. albicans*) was 338.4 ± 19.6 µg/disc versus 199.4 ± 52.7 µg/disc for the *L. plantarum* 14917-treated group. The addition of *L. plantarum* 14917 reduced the total protein in multispecies biofilms formed by *S. mutans* and *C. albicans* clinical isolates (from 3 independent experiments, n = 12, *p* < 0.001).

### 2.5. L. plantarum 14917 Altered 3D Structure of Biofilms Formed by S. mutans and C. albicans Clinical Isolates

Since *L. plantarum* 14917 demonstrated better inhibition of *C. albicans* and *S. mutans* isolates in planktonic and biofilm conditions, it was assessed for its impact on biofilm structure and mechanistic interaction. *L. plantarum* 14917 significantly reduced cariogenic biofilm formation, as measured by bacteria and EPS biomass, compared to the control group (*C. albicans–S. mutans* duo-species biofilm). The 72 h multispecies biofilms are shown in Figure 4A. In contrast to the complex and thick biofilms formed in the control group, the *L. plantarum* 14917-treated biofilms significantly reduced the biofilm thickness and biomass of both bacteria and EPSs (Figure 4B,C, *p* < 0.05). Furthermore, the horizontal coverage (Figure 4A) of the control group was also much broader than that of the treatment group, with nearly 60% bacterial coverage and 40 % EPS coverage in the most abundant layer (~30–40 µm above the substrate), while the treated group only had 20% coverage at the most abundant layer (~10 µm above the substrate). In addition, bacteria co-localized by EPSs were significantly fewer in number in the treatment group (Figure 4D, *p* < 0.05).

Furthermore, the added *L. plantarum* 14917 also significantly impacted the microcolony formation in the 72 h old multispecies biofilms. In contrast to the well-formed mushroom-shaped microcolonies in the control group, *L. plantarum* 14917-treated biofilms had significantly compromised microcolony structure (Figure 4A). Microcolonies are considered virulent and functional structures of biofilms. Surface-attached and free-floating microcolonies were quantified, and their numbers and sizes were compared between the control and *L. plantarum* 14917-treated biofilms. Numerous and large microcolonies were detected in the control group, while the intervention of *L. plantarum* 14917 resulted in fewer and smaller microcolonies (Table 2).

### 2.6. Plantaricin Inhibited the Growth of S. mutans and C. albicans Clinical Isolates

*S. mutans* (10^3^ CFU/mL) and *C. albicans* (10^1^ CFU/mL) were treated with plantaricin at a concentration ranging from 0 to 400 µg/mL and grown for 24 h in 1% glucose conditions. The MIC assay was carried out on three children’s *S. mutans* and *C. albicans* clinical isolates. Among the three pairs of *S. mutans* and *C. albicans* isolates, the MIC of plantaricin was 400 µg/mL for all three *S. mutans*, 200 µg/mL for two *C. albicans*, and 25 µg/mL for the other *C. albicans* strain. The clear culture was plated and incubated for an additional 48 h. Results reveal that the inhibitory effect on all these *S. mutans* and *C. albicans* isolates was bacteriostatic and fungistatic.

### 2.7. L. plantarum 14917 Downregulated C. albicans and S. mutans Virulence Genes in Biofilms

To assess the transcriptional regulations of *S. mutans* and *C. albicans* virulence genes in biofilms treated with *L. plantarum* 14917, qRT-PCR was performed for biofilms at 50 h, 2 h after the fresh culture medium was changed (and the sucrose concentration was changed from 0.1% to 1%). *S. mutans* genes related to EPS formation (*gtfB* and *gtfC*) and the *atpD* gene related to the assembled ATPase complex were significantly downregulated at 50 h. Meanwhile, genes related to *C. albicans* resistance fungal cell wall chitin remodeling (*CHT2*), resistance to antifungal medication (*ERG4*), and yeast hypha formation (*HWP1*) were also significantly downregulated following the culture medium change. The expressions of *S. mutans* (*gtfB*, *gtfC*, and *atpD*) and *C. albicans* (*ERG4*, *HWP1*, and *CHT2*) genes related to cariogenicity were downregulated by approximately 50–80% in the biofilms treated with *L. plantarum* 14917 for 50 h, compared to the control group (see Figure 5).

## 3. Discussion

Dysbiosis of microflora at various body sites has reported associations with disease conditions, including gastrointestinal diseases, dermatitis, and oral diseases, such as dental caries. Probiotics were initially introduced to the healthcare field to maintain or improve a healthy gut by modulating or restoring gut flora [22,23,24,25]. The usage of probiotics later expanded to many other medical fields [26], such as treating and preventing vaginal and urinary tract infections in women, controlling the development of food allergies or eczema in children, and stimulating and strengthening the human immune system [27].

Several studies have reported the inhibitory effect of probiotics on *S. mutans* and *C. albicans* in planktonic or single-species biofilm conditions [28,29,30,31]. The most commonly used probiotics include *Lactobacilli* and *Bifidobacterium*, both of which produce lactic acid and other bioactive substances, including hydrogen peroxide, carbon peroxide, diacetyl, low-molecular-weight antimicrobial substances, bacteriocins, and adhesion inhibitors that could potentially affect the growth of oral microorganisms [29]. For example, *Lactobacillus rhamnosus* GG, *Lactobacillus reuteri*, *Lactobacillus casei*, *Lactobacillus plantarum*, and *Lactobacillus salivarius* inhibit the growth of *S. mutans* in vitro and in vivo [32,33,34]. Moreover, *L. reuteri*, *L. rhamnosus*, *L. casei*, *L. paracasei*, *L. fermentum*, and *L. acidophilus* inhibit the growth of *C. albicans* in vitro and in vivo [35,36,37]. Since the co-existence of *S. mutans* and *C. albicans* in the oral cavity leads to more pathogenic microbial eco-community and potentially elevates the caries risk of individuals, an ideal probiotic regimen is one that controls *S. mutans* and *C. albicans* simultaneously. Limited studies, however, have assessed the effect of probiotics on *S. mutans* and *C. albicans* in a multispecies setting [14,19,20].

### 3.1. Susceptibility of Clinically Isolated S. mutans and C. albicans to Common Antimicrobial Agents

Studies from various researchers have shown that clinically isolated bacteria and yeast have a higher tolerance to antimicrobial reagents, such as antibiotics and antifungal medications; this developed resistance poses a challenge in bacteria and yeast control in clinical settings and requires a better antimicrobial regimen. Similar phenomena have been seen for *S. mutans*; for example, the study reported by AL-Shami et al. [38] showed significant levels of penicillin, erythromycin, amoxicillin, clindamycin, and lincomycin resistance in *S. mutans* clinical isolates in dental patients. Similarly, Pasquantonio et al. [39] reported that 14% of 50 clinically isolated *S. mutans* had resistance to penicillin, although quinolones and rifampicin were confirmed to have an excellent activity against oral streptococci.

Clinically isolated *C. albicans* have also shown less susceptibility to antifungal medication. Jahanshiri et al. reported that the levels of resistance of 32 *C. albicans* isolates from head and neck cancer patients to fluconazole, itraconazole, ketoconazole, and amphotericin B were 62.5%, 81.25%, 93.75%, and 87.50%, respectively [40]. Aitken-Saavedra J et al. tested the susceptibility of *C. albicans* clinical strains of type II diabetic patients to fluconazole therapy and nystatin [41]. At higher pH, *C. albicans* yeast is less susceptible to nystatin, whereas at lower pH, *C. albicans* is more susceptible to treatment, suggesting that it might have greater effectiveness in patients with higher salivary acidification. Fluconazole had no significant results in their study. Biofilms are considered to provide a protective environment that enhances *C. albicans*’ resistance to antifungal medications. CHANDRA et al. reported that *C. albicans* cells that underwent 72 h of biofilm treatment were highly resistant against amphotericin B, fluconazole, nystatin, and chlorhexidine in an in vitro model, and this increased virulence is similar to what occurs in vivo [42]. Furthermore, topical antifungal agents such as fluconazole have negligible effects on *C. albicans* in mixed oral biofilms, due to the protective microbial exopolysaccharide matrix. However, antifungal susceptibility was enhanced by introducing povidone iodine, which inhibited exopolysaccharide matrix formation, resulting in a reduction in *C. albicans* [43].

Since clinically isolated bacteria and yeast were more resistant to antimicrobial reagents, it is possible that our experimental isolates were in the same situation. We want to emphasize that, while clinically isolated *S. mutans* and *C. albicans* may be more resistant to antimicrobial reagents, the tested probiotics significantly inhibited the growth of *S. mutans* and *C. albicans* isolates in both planktonic and biofilm conditions.

### 3.2. Equal Effectiveness of L. plantarum 14917 on Clinical Isoalted C. albicans and S. mutans Compared to Wild-Type Strains

Our previous work [21] assessed the inhibition of probiotic *Lactobacillus* spp. on the growth and biofilm formation of wild-type *C. albicans* and *S. mutans.* The results of the current study indicate that the inhibitory effect of *L. plantarum* on clinically isolated *C. albicans* and *S. mutans* is similar to its effect on the wild-type strains. Of note is that the inoculation concentration of *S. mutans* we used in the current study (10^6^ CFU/mL) was even 1-log higher than that used in our previous wild-type study (10^5^ CFU/mL), which demonstrated a higher challenge for *L. plantarum* during the inhibitory interactions. In terms of the change in the pH level in the planktonic and the culture medium of the biofilm model over time, no significant differences were noted between the current study, which used clinical isolates, and the previous study, which used *C. albicans* and *S. mutans* laboratory strains.

Regarding the effectiveness of growth inhibition in the planktonic model, the findings of this study are also similar to those of the study of Zeng et al., in which *L. plantarum* 14917 demonstrated significant inhibition of *C. albicans* by 1-log for at six hours, and 2-log at 20 h. *S. mutans* laboratory and clinical strains were significantly inhibited by 2-log at 6 h, and completely inhibited at 20 h for the clinical isolate in the planktonic model. Both studies showed comparable and significant inhibition of the cariogenic biofilm at 72 h (1% sucrose condition) in terms of the morphological structure of the biofilm, dry weight, biomass of bacteria and EPSs, bacteria co-localization by EPS, and the number, size, and surface area of microcolonies. The mechanistic assessment indicated that *L. plantarum* 14917 inhibited the expression of *S. mutans* (*gtfB*, *gtfC*, and *atpD*) and *C. albicans* (*ERG4*, *HWP1*, *and CHT2*) virulence genes and virulent structures, such as *C. albicans* hypha formation, which is consistent with our previous findings on wild-type *C. albicans* and *S. mutans* [21].

*L. plantarum* 14917 demonstrated equal effectiveness in inhibiting clinically isolated *C. albicans* and *S. mutans* from S-ECC children compared to wild-type strains, indicating *L. plantarum* 14917’s strong potential to be incorporated into a future clinical caries prevention and control regimen to target cariogenic pathogens.

Despite the promising finding that *L. plantarum* inhibited the growth of *C. albicans* and *S. mutans* clinical isolates and their biofilm formation, we recognize the study’s limitations. Although clinical isolates of *C. albicans* and *S. mutans* were used, the study was conducted using an in vitro model; further animal and clinical studies are required to verify their safety and efficacy in an oral biofilm complex, as well as to observe long-term oral microbial virulence.

## 4. Materials and Methods

### 4.1. Study Design

This study was designed in six steps to screen the best-performing probiotic *Lactobacillus* spp. on *S. mutans* and *C. albicans* clinical isolates. The study scheme is shown in Appendix A. In Step 1, the inhibitory effect of *Lactobacilli* was assessed in the planktonic condition against *C. albicans* and *S. mutans* isolates from two S-ECC children. The best-performing *Lactobacillus* advanced to Step 2 to verify its inhibitory effect against *C. albicans* and *S. mutans* isolates from an additional eight S-ECC children. In Step 3, two of the three *Lactobacilli* with a higher inhibition of *S. mutans* and *C. albicans* in the planktonic condition were further tested with *C. albicans* and *S. mutans* isolated from two S-ECC children in a multispecies biofilm condition. In Step 4, the better-performing *Lactobacillus* advanced to Step 5 to assess its effect on cariogenic biofilm structure. Moreover, molecular assays were used to assess the mechanistic interactions between *Lactobacillus*, *S. mutans*, and *C. albicans* in biofilms in Step 5. Lastly, the effect of plantaricin, an antimicrobial peptide produced by *L. plantarum*, on the growth of *S. mutans* and *C. albicans* was examined in Step 6.

### 4.2. Bacterial Strains and Starter Preparation

The microorganisms used in this study were *L. plantarum* ATCC 8014, *L. plantarum* ATCC 14917, *L. salivarius* ATCC 11741, the pairs of *S. mutans* and *C. albicans* isolated from ten preschool children with S-ECC. The clinical isolates were archived from a previous study [44]. The current study did not involve human subject interaction.

*Lactobacillus*, *S. mutans*, and *C. albicans* were recovered from frozen stock using MRS agar (BD Difco™, 288210), blood agar (TSA with Sheep Blood, Thermo Scientific™ R01202, Wilmington, DE, USA), and YPD agar (BD Difco™, 242720) for 48 h, respectively. Altogether, 3–5 colonies of each species were inoculated in 10 mL of broth for overnight incubation (5% CO_2_, 37 °C). *Lactobacillus* spp. were grown in MRS broth (BD Difco™, 288130); *S. mutans* was grown in TSBYE broth (3% Tryptic Soy, 0.5% Yeast Extract Broth, BD Bacto™ 286220 and Gibco™ 212750) with 1% (*w*/*v*) glucose; and *C. albicans* was grown in YPD broth (BD Difco™, 242820). On the following day, 0.5 mL of the overnight starters was added to individual glass tubes with fresh broth and incubated for 3–4 h to reach the mid-exponential phase with desirable optical density (OD). The morning starters were then ready for the preparation of planktonic and biofilm models in this study.

### 4.3. Planktonic Model

Interactions between *S. mutans*, *C. albicans*, and *Lactobacillus* species were first evaluated in planktonic conditions using the method established in our previous study [21]. Briefly, the inoculation quantity of *S. mutans* (10^5^ CFU/mL) and *C. albicans* (10^3^ CFU/mL) was chosen for simulating high-caries-risk conditions in the clinical setting; the inoculation quantity of *Lactobacillus* (10^8^ CFU/mL) was used for inhibiting the growth of *S. mutans* and *C. albicans*. A pair of *S. mutans* and *C. albicans* obtained from the same child and one of the *Lactobacilli* were inoculated in 10 mL of TSBYE broth with 1% (*w*/*v*) glucose and incubated for 20 h (5% CO_2_, 37 °C). The growth of each microorganism was assessed using blood agar at 0, 6, and 20 h. Appendix A indicates distinct morphological differences between *S. mutans*, *C. albicans*, and *Lactobacillus* spp. The pH values were measured at 0, 2, 4, 6, and 20 h.

### 4.4. Mixed-Species Biofilm Model

A mixed-species biofilm model was used to assess the effect of *Lactobacilli* on the biofilm formation of *S. mutans* and *C. albicans*. The biofilm method involved using saliva-coated hydroxyapatite discs (0.50″ diameter × 0.05″ thickness, Clarkson Chromatography Products, Inc., South Williamsport, PA) following the concept of “ecological plaque-biofilm” [45]. The discs were placed in a vertical position using a custom-made disc holder to mimic the caries-prone smooth tooth surfaces in the oral cavity [9,21,46,47,48].

The mixture of *S. mutans*, *C. albicans*, and *Lactobacilli* was inoculated in 2.8 mL of TSBYE broth with 0.1% (*w*/*v*) sucrose, and incubated at 37 °C and 5% CO_2_. The organisms were grown for the first 24 h to allow the initial biofilm to form. At 24 h of biofilm growth, the discs with biofilms were transferred to a fresh culture medium containing 1% (*w*/*v*) sucrose to induce cariogenic challenges. The culture medium was replaced once every 24 h until the end of the experimental period (at 72-h), and *Lactobacilli* (10^8^ CFU/mL) were added to the fresh culture medium daily during the replacement for treatment groups. The pH of the culture medium was measured at selected time points using a standard pH electrode. The biofilms were processed for microbiological properties, dry weight, confocal imaging assays, and qRT-PCR at specific time points, according to the study design, to assess the regulation of *Lactobacilli* on the expression of *C. albicans* and *S. mutans* virulence genes.

### 4.5. Microbiological Analysis of the Mixed-Species Bacterial Population

The biofilms were homogenized by sonication, as detailed previously [21,48,49,50]. The homogenized suspension was used to determine the number of viable cells by plating on blood agar using an automated EddyJet Spiral Plater (IUL, SA, Barcelona, Spain). Three species were differentiated by colony morphology in conjunction with microscopic examination of cells from selected colonies [51]. The homogenized suspension was also used for the measurement of dry weight, as detailed previously [21,48,50]. Duplicated discs were used in each run. Independent assays were repeated three times.

### 4.6. Laser Scanning Confocal Fluorescence Microscopy (LCSFM) Imaging of Biofilm Matrix

We assessed two essential components of the biofilm matrix, bacteria and exopolysaccharides (EPSs), using LCSFM, as detailed previously [21,48,50]. Briefly, 1 μM Alexa Fluor^®^ 647-labeled dextran conjugate (Molecular Probes, Invitrogen Corp., Carlsbad, CA, USA) was added to the culture medium at the beginning and during the development of the biofilms for exopolysaccharide visualization. The bacterial species and fungal species were labeled with SYTO^®^ 9 green fluorescent nucleic acid stain (485/498 nm; Molecular Probes). The images were obtained using an Olympus FV 1000 two-photon laser scanning microscope (Olympus, Tokyo, Japan) equipped with a 10× (0.45 numerical aperture) water immersion objective lens. Each biofilm formed on the HA disc was scanned at 5 positions randomly [52]. Twenty image stacks were collected for each experimental condition.

### 4.7. Computational Analyses of the Confocal Biofilm Images

To visualize the morphology and 3D architecture of the biofilms, each structural component (EPSs and bacteria) was rendered in 3D using Amira 5.0.2 (Mercury Computer Systems Inc., Chelmsford, MS, USA) [52,53]. For biofilm quantitative analysis, COMSTAT and DUOSTAT (http://www.imageanalysis.dk, accessed on 1 January 2020.) were employed. We calculated the biomass, quantity, and dimensions (volume, diameter, and height) of microcolonies using the software COMSTAT, and we utilized DUOSTAT to determine the co-localization of two biofilm components (EPSs and bacterial cells, for example) across the 3D biofilm architecture [48].

### 4.8. Assessment of Total Protein in Multispecies Biofilms

The biofilms were homogenized via sonication, as detailed previously [21,48,49,50]. The 48 h old biofilms were harvested from the 1 HA disc in 1 mL of 0.89% *w*/*w* saline. A total of 150 µL of homogenized suspension was used for total protein assessment using a Pierce™ BCA Protein Assay Kit (Thermo Scientific™, Wilmington, DE, USA) with a microplate reader (Tecan Infinite^®^ 200 PRO, Männedorf, Switzerland). Three independent experiments were conducted for each condition.

### 4.9. Inhibition of S. mutans and C. albicans by Plantaricin

Bacteriocins, antimicrobial molecules produced by *L. plantarum*, are also known as plantaricins [54]. Peptide plantaricin-149 (acetate) powers (Creative Peptides, Shirley, NY, USA) were dissolved in ddH_2_O to prepare plantaricin solutions. *S. mutans* (10^3^ CFU/mL) and *C. albicans* (10^1^ CFU/mL) from 3 S-ECC children were selected and treated with the plantaricin at a concentration of (0–400 µg/mL). The mixtures of plantaricin with *S. mutans* or *C. albicans* were grown for 24 h in TSBYE with 1% glucose in 96-well plates. Clear culture after 24 hours’ incubation indicated no growth of microorganisms. Therefore, the minimal inhibition concentration (MIC) of plantaricin-149 was defined as the lowest concentration that inhibited the growth of *S. mutans* and *C. albicans*.

### 4.10. Real-Time Reverse Transcription Polymerase Chain Reaction (qRT-PCR)

The mass of biofilms was harvested from four discs at 50 h for each condition. The discs were immersed in RNALater (Applied Biosystems/Ambion, Austin, TX, USA) for 1 h, followed by biomass removal with a spatula. RNAs were extracted and purified with a MasterPure complete DNA and RNA purification kit (epicenter, Lucigen, WI, USA). Raw RNA product was quantified using a NanoDrop One Microvolume UV-Vis Spectrophotometer (Thermo Scientific™, Wilmington, DE, USA). A 2nd purification was performed with DNase I (Thermo Fisher Scientific, Invitrogen™ 18068015, Wilmington, DE, USA).

Then, cDNAs were synthesized using 0.4 μg of purified RNA and the BioRad iScript cDNA synthesis kit (Bio-Rad Laboratories, Inc., Hercules, CA, USA). The resulting cDNAs and negative controls were amplified by quantitative amplification using Applied Biosystems™ PowerTrack™ SYBR Green Master Mix and a QuantStudio™ 3 Real-Time PCR System (Thermo Fisher Scientific, Wilmington, DE, USA). Each 20 µL reaction mixture included template cDNA, 10 µM of each primer, and a 2× SYBR-Green mix (containing SYBR-Green and Taq DNA Polymerase). Unique core genes of *S. mutans* and *C. albicans* were used as an internal reference for comparative expression calculation: *gyrA* for *S. mutans* genes [55] and *ACT1* for *C. albicans*. Expression of virulence genes of *S. mutans* (*gtfB*, *gtfC*, and *atpD*) and *C. albicans* (*CHT2*, *HWP1*, and *ERG4*) was assessed using the methods detailed previously [21].

### 4.11. Assessment of Morphology of C. albicans

The 24 h old planktonic culture medium was observed under a light microscope (Olympus BX43, 214, Tokyo, Japan) with 100X oil objective (Olympus UPlanFL N 100X, Tokyo, Japan) to assess the morphological changes in *C. albicans*. A total of 20 µL of culture medium was placed on the glass slide and visualized without staining.

### 4.12. Statistical Analysis

To compare the abundance of *S. mutans*, *C. albicans*, and *Lactobacillus* spp. in planktonic and biofilms, the CFU values were first converted to natural log values. Zero values of the CFU remained at zero. The natural log values were compared between groups using the t-test for two groups or ANOVA for more than two groups, after assessing the normality of data. For other parameters, such as biomass (bacteria and EPS), the number and size of microcolonies, and the pH value of the biofilms at specific time points, normality tests were performed first. For normally distributed data, the comparisons between groups were tested using the *t*-test for two groups and one-way ANOVA for more than two groups followed by the post hoc test. For non-normally distributed data, the Kruskal–Wallis test was used to compare the outcomes of more than two groups, and Mann–Whitney tests were used for the comparison of two groups. Statistical tests were two-sided with a significant level of 5%. IBM SPSS was used for statistical analyses.

### 4.13. Ethical Approval

Ethical approval of the study and the written consent/permission forms were obtained from the Research Subject Review Board at the University of Rochester (RSRB00056870, date of approval: 17 May 2015).

## 5. Conclusions

Our study findings show a remarkable inhibitory effect of *L. plantarum* 14917 on *S. mutans* and *C. albicans* clinical isolates, resulting in a reduced biofilm structure with a significantly smaller microbial and extracellular matrix and less virulent microcolonies structure. The mechanistic assessment indicated that *L. plantarum* 14917 had a positive inhibitory impact on the expression of *S. mutans* and *C. albicans* virulence genes and virulent structures, such as *C. albicans* hypha formation. Future utilization of *L. plantarum* 14917 and/or its antimicrobial peptide plantaricin could lead to a new paradigm shift in dental caries prevention.

## Figures and Tables

**Figure 1 ijms-24-02991-f001:**
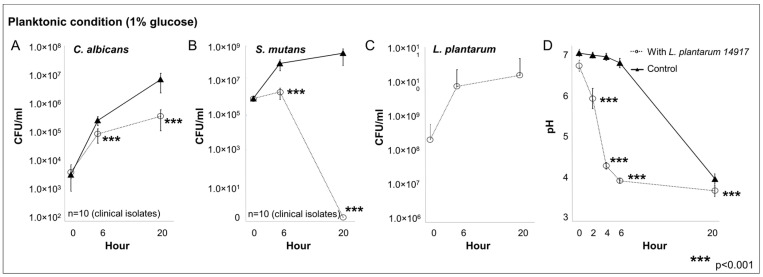
Inhibitory effect of *L. plantarum* on clinically isolated *C. albicans* and *S. mutans* from children with early childhood caries in multispecies planktonic condition. *C. albicans* and *S. mutans* clinical strains were isolated from 10 children with early childhood caries (ECC). Experiments were repeated in triplicate. Each planktonic multispecies condition included the *C. albicans* and *S. mutans* isolated from the same ECC child, with/without added *L. plantarum* 14917. The control group only consisted of *C. albicans* and *S. mutans*. (**A**) *L. plantarum* inhibited the growth of *C. albicans*. (**B**) *L. plantarum* significantly inhibited the growth of *S. mutans*. (**C**) *L. plantarum* maintained a stable growth during the 20 h. (**D**) The culture medium pH.

**Figure 2 ijms-24-02991-f002:**
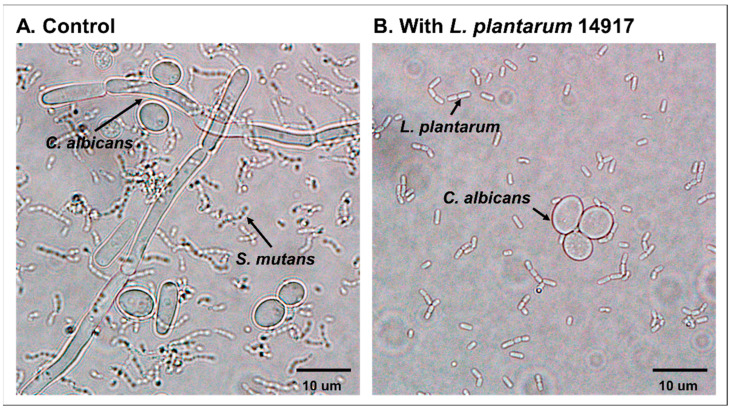
Inhibition of *C. albicans* hypha formation by *L. plantarum* 14917. (**A**) *S. mutans* and *C. albicans* grown in 1% glucose at 24 h. (**B**) *S. mutans* and *C. albicans* grown in 1% glucose with added *L. plantarum* 14917 at 24 h.

**Figure 3 ijms-24-02991-f003:**
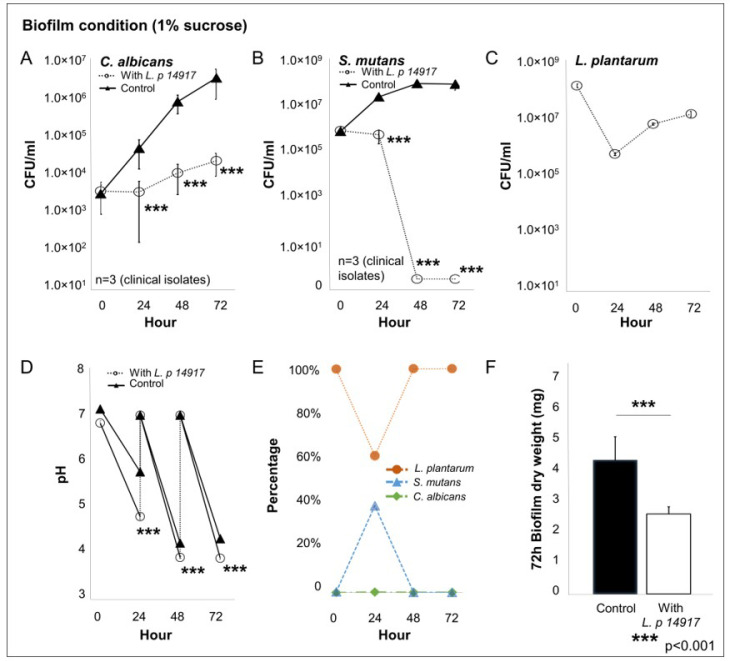
Interaction of *L. plantarum* 14917 and clinically isolated *C. albicans* and *S. mutans* in multispecies biofilms. Multispecies biofilms were formed by *L. plantarum* 14917 and clinically isolated *C. albicans* and *S. mutans* from three children with ECC. The treated group was grown with added *L. plantarum* 14917. (**A**–**C**) The growth of *C. albicans*, *S. mutans*, and *L. plantarum* 14917 in multispecies biofilms is plotted. (**D**) The pH of the culture medium. (**E**) The composition of each microorganism. (**F**) The dry weight of biofilms at 72 h.

**Figure 4 ijms-24-02991-f004:**
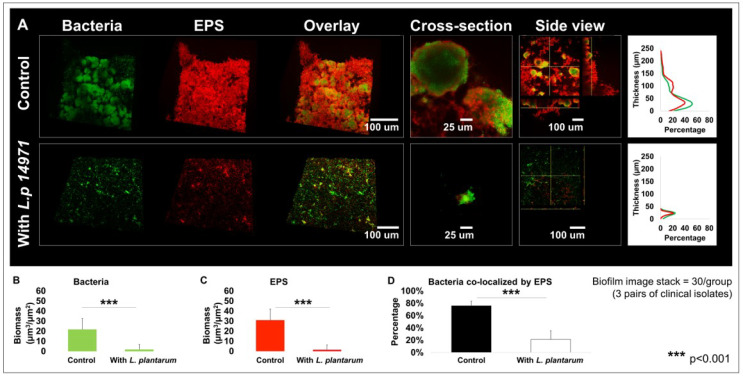
Changes in multispecies biofilm 3D structure caused by *L. plantarum* 14917. Biofilms were formed by *C. albicans* and *S. mutans* only (control) and treated with *L. plantarum* 14917 in 1% sucrose condition and visualized via two-photon laser confocal microscope at 72 h. Amira software was used to reconstruct the images as 3D structures, to visualize bacterial channels (green), EPS channels (red), and the bacteria/EPS overlay. (**A**) Biofilm structure. The layer distribution of the biofilms indicating that the biofilms grown in the control group were much thicker than those of the treatment group. (**B**) Biomass formed by bacteria. (**C**) Biomass formed by EPSs. (**D**) Bacteria co-localized by EPSs. Biofilm parameters were calculated using data from three biofilms formed by *C. albicans* and *S. mutans* isolated from three ECC children. For each biofilm, 10 randomly selected points of biofilms were visualized.

**Figure 5 ijms-24-02991-f005:**
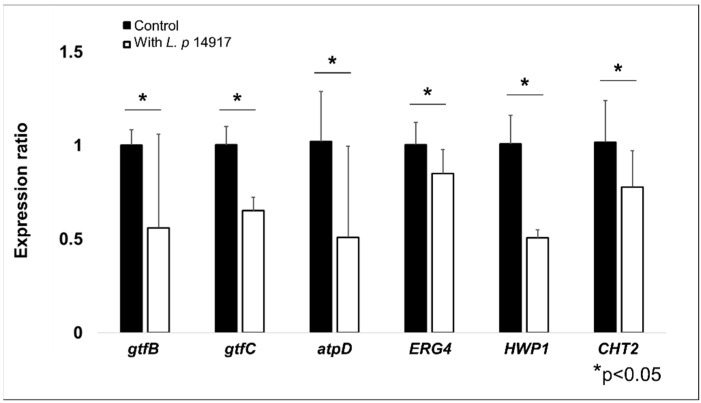
Regulation of *S. mutans* and *C. albicans* virulence genes by *L. plantarum* 14917 in multispecies biofilms. The expression of *S. mutans* (*gtfB*, *gtfC*, and *atpD*) and *C. albicans* (*ERG4*, *HWP1*, and *CHT2*) genes related to carcinogenicity were reduced in the biofilms treated with *L. plantarum* 14917 for 50 h, compared to the control group (*S. mutans* and *C. albicans*). * *p* < 0.05 when comparing the control and *L. plantarum* 14917-treated biofilms.

**Table 1 ijms-24-02991-t001:** Demographics data and characteristics of study participants (n = 10).

Items	Mean (SD) or %
Age (Year)	3.5 ± 1.0
Gender (Male)	50%
Race: Caucasian	60%
African American	30%
Asian	10%
Ethnicity: Hispanic	10%
Brushing Frequency (Daily)	90%
Attending Daycare (Yes)	20%
Plaque Index	1.8 ± 0.6
*dt*	11.7 ± 5.1
*mt*	0.2 ± 0.6
*ft*	0.1 ± 0.3
*dmft*	12 ± 4.9
*ds*	27.2 ± 17.4
*ms*	1.0 ± 3.2
*fs*	0.1 ± 0.3
*dmfs*	28.3 ± 16.5

Note: clinical isolates were obtained from the children with ECC. *dt*: decayed teeth. *mt*: missing teeth. *ft*: filled teeth. *dmft*: decayed/missing/filled teeth. *ds*: decayed surfaces. *ms*: missing surfaces. *fs*: filled surfaces. *dmfs*: decayed/missing/filled surfaces.

**Table 2 ijms-24-02991-t002:** Quantitative assessment of microcolonies in multispecies biofilms.

Microcolony Parameters	Control (n = 30)	With *L. plantarum* 14917 (n = 30)
Number of attached microcolonies	15.3 ± 4.1	9.2 ± 6.7 **
Area of attached microcolonies (um^2^)	972.2 ± 924.4	283.6 ± 84.3 **
Volume of attached microcolonies (um^3^) × 10^3^	3471.5 ± 2334.8	25.6 ± 27.5 ***
Number of free microcolonies	287.4 ± 71.1	213.3 ± 77.5 ***
Diameter of free microcolonies (um)	38.0 ± 4.3	28.4 ± 3.0 ***
Volume of free microcolonies (um^3^) × 10^3^	925.0 ± 255.0	145.8 ± 111.8 ***

Note: ** *p* < 0.01, *** *p* < 0.001 when comparing the control and *L. plantarum* 14917-treated biofilms. n = 30 indicates that 30 biofilm image stacks were used for each group (control and treatment).

## Data Availability

All data generated or analyzed during this study are included in this article. Further enquiries can be directed to the corresponding author.

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
