# Peer review of "Effect of Probiotic Lactobacillus plantarum on Streptococcus mutans and Candida albicans Clinical Isolates from Children with Early Childhood Caries"

_ijms, 2023, doi:10.3390/ijms24032991_

Round 1
Reviewer 1 Report
Zeng et. al., presented a very nice piece of work where they showed the inhibitory effect of L. plantarum 14917 against S. mutans and C. albicans growth and their virulent factors.
1. I suggest quantifying the total biofilm proteins formed by the Streptococcus mutans and Candida albicans before and after treatment with probiotics.
2. In the abstract portion, first line needs to be modified as emerging pathogenic bacteria/fungi for dental caries.
Author Response
Dear reviewer,
Thank you for your insightful review and helpful comments that have enabled us to correct errors and improve our manuscript. We have addressed your comments, concerns, and suggestions to the best of our ability. Material changes in the manuscript are highlighted in yellow or using “Track changes” for your convenience in reviewing the changes.
- I suggest quantifying the total biofilm proteins formed by the Streptococcus mutans and Candida albicans before and after treatment with probiotics.
Response: We have added the total biofilm proteins quantification data in the revised manuscript. The addition of L. plantarum 14917 reduced the total protein in multispecies biofilms.
- In the abstract portion, first line needs to be modified as emerging pathogenic bacteria/fungi for dental caries.
Response: We revised this point in the revision.
Reviewer 2 Report
The manuscript "Effect of probiotic Lactobacillus plantarum on Streptococcus mutans and Candida albicans clinical isolates from children with early childhood caries" deals with an interesting topic of the influence of probiotic cells on pathogens. However, in my opinion, the current work is only a supplement of Authors' previous work from 2022 "Lactobacillus plantarum Disrupts S. mutans-C. albicans Cross-Kingdom Biofilms", where they showed a similar effect on laboratory isolates as those obtained for clinical strains. Authors conclude that the strains isolated from patients may be more resistant to drugs, but for the studied strains, they do not have any research on this issue and no evidence of how the strains differ. However, the conclusions for both types of strains are quite similar. Moreover, the previous work also contained more advanced proteomic analyses that could be interesting to compare. The lack of a clear novelty aspect is a major shortcoming of this work, but I also have other comments.
Only one probiotic species is mentioned in the title, and the authors are studying two. The title should be changed to just the genus name or to both species.
There is a serious mistake in the abstract when Candida is defined as a bacterium instead of a fungus.
The sentences in the Introduction in lines 36-39 are unnecessary.
The abbreviations in Table 1 are not explained.
In the Results section, the captions under the figures are a repetition of the main text. They should be more of a summarized methodology than a description of the results. And in the main text there is a lack of reference to the Fig. 1 panels when describing the results.
Supplementary results about the comparison of probiotic strains should be in the main text of the manuscript.
Why are biofilm results done for three isolates and MFC determination for plantaricin only for one strain? This is definitely less reliable, especially for MFC determination. Why, for the latter, are individual results not presented, and this experiment could be performed additionally with the counting of CFUs. However, another issue is that differentiation between colonies of microbial species growing on the same agar plates is questionable, especially when antibiotic/antimycotic and differential media are available for use.
The results with Fig. 5 should be presented earlier, after Fig. 1. as they concern the effect of bacteria on filamentation, which is preceding to biofilm formation.
Is it known what is the level of plantaricin production by the tested strains of bacteria?
Section 3.1 in the Discussion is redundant because it is not known what the characteristics of the tested strains are. Perhaps the authors should test the synergism/antagonism of plantaricin with selected antifungals and antibiotics.
Minor:
line 23 –species name with italics
line 49 – fungi without italics
line 55 – capital letter for Lactobacillus
line 111 – spp without italics
line 127 – strain number without italics
Round 2
Reviewer 2 Report
The authors have clarified most of the comments, they may include the figure with different colonies in the supplementary material, and should complete and explain their rationale in the text as to why they want to keep the section 3.1 in the discussion.
Author Response
Dear reviewer,
Thank you again for your insightful review and helpful comments that have enabled us to correct errors and improve our manuscript. We have addressed your comments and suggestions to the best of our ability. Material changes in the manuscript are highlighted in yellow or using “Track changes” for your convenience in reviewing the changes.
Reviewer 2:
The authors have clarified most of the comments, they may include the figure with different colonies in the supplementary material, and should complete and explain their rationale in the text as to why they want to keep the section 3.1 in the discussion.
Response: We really appreciate the reviewer’s comments and suggestions! We added Figure S5 “Distinct morphological differences between S. mutans, C. albicans, and Lactobacillus spp.” in the supplementary material. In addition, we also added summative text at the end of Discussion section 3.1 (Lines 284 to 288), to emphasize the importance and the rationale to keep section 3.1.